# CO_2_ Capture Performance and Preliminary Mechanistic Analysis of a Phase Change Absorbent

**DOI:** 10.3390/molecules30163404

**Published:** 2025-08-18

**Authors:** Chuanyong Zhu, Yucai Zhang, Baoyue Zhang, Chongqing Xu, Guihuan Yan, Na Yang

**Affiliations:** 1School of Environmental Science and Engineering, Qilu University of Technology (Shandong Academy of Sciences), Jinan 250353, China; zchychina@163.com (C.Z.); z1641233898@163.com (Y.Z.); zhangbaoyue15@163.com (B.Z.); xucq@sderi.cn (C.X.); yanguihuan@163.com (G.Y.); 2Ecology Institute of Shandong Academy of Sciences, Qilu University of Technology (Shandong Academy of Sciences), Jinan 250103, China; 3Shandong Technology Innovation Center of Carbon Neutrality, Qilu University of Technology (Shandong Academy of Sciences), Jinan 250103, China

**Keywords:** CO_2_ capture, functionalized ionic liquids, phase change absorbent, reaction mechanism, regeneration energy consumption

## Abstract

Phase change absorbents are deemed a promising alternative for CO_2_ capture due to their excellent CO_2_ absorption performance, good stability, and low renewable energy consumption. To address the issues of insufficient loading capacity, low regeneration efficiency, and high energy consumption during regeneration in current chemical absorbents, a novel phase change absorbent was developed. As an amino acid ionic liquid phase change absorbent with tetraethylenepentamine as the cation, imidazole as the anion, and n-propanol as the phase separation agent, this absorbent offers a potential solution. The highest absorption capacity of the [TEPAH][Im]/NPA/H_2_O system at the optimal n-propanol-H_2_O ratio (1:1) reaches 1.34 mol·mol^−1^, and the viscosity of the CO_2_-rich phase amounts to a mere 3.58 mPa·s. Additionally, the desorption efficiency reached 91.1% at 363.15 K, while the loading capacity in the fifth cycle remained over 1.16 mol·mol^−1^. As n-propanol is present in the [TEPAH][Im]/NPA/H_2_O system, the rich phase makes up roughly 30% of the total volume. The energy consumption for regeneration of the [TEPAH][Im]/NPA/H_2_O phase change absorption system is 2.20 GJ·t^−1^ CO_2_. Under identical regeneration conditions, the system can reduce the regeneration energy consumption by 41.6%.

## 1. Introduction

As the most significant greenhouse gas, carbon dioxide (CO_2_) emissions account for approximately 82% of total greenhouse gas emissions [1]. Reducing these emissions is essential to mitigate global warming and prevent climate deterioration. CO_2_ capture, utilization, and storage (CCUS) is regarded as a critical technology for curbing near-term CO_2_ emissions and has been extensively deployed in industrial settings [2]. This procedure entails capturing carbon dioxide emissions from industrial exhaust streams through advanced separation technologies, subsequently pressurizing and conveying the purified CO_2_ for either industrial repurposing or permanent geological sequestration. CCUS consists of four primary components—capture, transport, utilization, and storage—among which CO_2_ capture is considered the most crucial element of the entire process [3]. CO_2_ capture technologies primarily include pre-combustion capture, oxygen-rich combustion capture, and post-combustion capture [4,5]. The most established method is traditional absorption using organic amine solutions [6]. Nonetheless, this method encounters difficulties; specifically, the process of regenerating the chemical absorbents utilized in carbon capture is characterized by significant energy consumption. In response to these challenges, phase change absorption technology has emerged [7,8,9]. By adjusting load or temperature, this innovative method induces liquid–liquid or solid–liquid phase changes during CO_2_ absorption, leading to CO_2_ accumulation in one phase. Only the enriched phase requires desorption, significantly reducing the heat consumption involved in regenerating the CO_2_ absorbent [10,11,12]. Currently, most phase change absorbents are based on amine reagents. While these absorbents can effectively reduce the energy consumption associated with CO_2_ capture, reports indicate that amine-based phase change absorbents have the drawback of high viscosity in the CO_2_-rich solution. This increased viscosity can exacerbate equipment corrosion and result in higher transportation costs [13,14,15].

Among many new absorbers, phase change absorbers are a type of CO_2_ absorber with great energy-saving potential; they can transition from a homogeneous phase to liquid–liquid or liquid–solid phases through temperature and load control, and most of the CO_2_ is enriched in one of the phases. Therefore, this absorber can reduce the energy consumption of CO_2_ regeneration by decrease the amount of liquid requiring regeneration. However, this type of phase change absorbent primarily relies on physical absorption, which results in lower absorption efficiency [16]. Amino ionic compound phase change absorbents can enhance absorption performance by increasing the number of amino groups. However, while these products exhibit high structural stability, their regeneration performance requires improvement [17]. In addition, the existing ionic compound phase change system is a terpolymer phase change system composed of ionic compound/ethanol/water [18]. Although this system uses ethanol as a phase separation agent to enhance phase separation due to its high vapor pressure, the organic solvent’s low boiling point results in substantial solvent loss during pyrolysis, ultimately raising renewable energy requirements [19,20,21]. To further enhance the performance of phase change absorbent and reduce renewable energy consumption, it is imperative to devise a new-generation phase change absorbent with higher energy-saving capabilities and greater effectiveness [22].

The objective of this research was to design a novel liquid–liquid biphasic absorbent comprising ionic compound, n-propanol(NPA), and water. In this new type of phase change absorbent, the main absorbent is selected as tetraethylenepentamine (TEPAH) and imidazole (Im) containing active groups, which not only ensures the absorption load and absorption capacity, but also promotes the hydrolysis conversion of carbamate, thereby breaking through the defect of low regeneration efficiency. In addition, choosing NPA with low vapor pressure as the phase separator can not only enhance the regeneration performance but also solve the problem of large solvent loss during pyrolysis; finally, water serves as a solvent to decrease the viscosity of the rich-phase absorbent. In this study, the absorption–desorption properties, the proposed mechanism of phase transition and regeneration energy consumption of ionic compound phase change absorbents were investigated, which could establish a theoretical groundwork for advancing the industrial deployment of phase change absorbents.

## 2. Results and Discussion

### 2.1. CO_2_ Absorption Performance of [TEPAH][Im]/NPA/H_2_O

#### 2.1.1. Optimization of the Component Ratio for CO_2_ Absorption by [TEPAH][Im]/NPA/H_2_O

The CO_2_ absorption properties of the [TEPAH][Im]/NPA/H_2_O at 303.15 K were investigated. The phenomenon of phase separation is depicted in Figure 1. Previous research [23] indicated that the NPA-H_2_O volume ratio affected the absorption capacity, volume fraction of the rich phase, and viscosity of the solution. In order to find the best ratio, the CO_2_ absorption performance of [TEPAH][Im]/NPA/H_2_O under different solvent volume ratios was analyzed using the absorption process in Section 3.3. As illustrated in Figure 1, when the volume ratio of NPA-H_2_O reaches 1:1, the absorbent forms two distinct liquid phases after absorption saturation. The main components of the upper phase are [TEPAH]^+^, with a small amount of [Im]^−^ and NPA, while the main components of the lower phase are carbonate/bicarbonate (HCO_3_^−^/CO_3_^2−^), RNCOO^−^, and CO_2_. This is mainly because the solubility of the absorption product and NPA in water is different, and the absorption product is basically insoluble in NPA but can dissolve well in water. When the water content in the system is relatively high, NPA cannot be completely dissolved, but it can dissolve and absorb the product well. Therefore, the system exhibits favorable liquid–liquid phase change behavior. As shown in Figure 2, as the volume ratio of NPA-H_2_O increases, the volume of the lower phase decreases, while the volume of the upper phase increases following saturation by adjusting the volume fraction of NPA in the absorbent solution; the dissolution of CO_2_ in the phase change solvent can be modulated, thereby enabling control over the phase transition behavior. As Table 1 distinctly indicates, the viscosity of the fresh absorbent increases with higher NPA-H_2_O volume ratios due to the inherently greater viscosity of NPA compared to water. In the absence of phase separation after absorption saturation, the viscosity of the absorbent slightly increases compared to the fresh solution. After absorbing carbon dioxide, the viscosity of the lower phase increases compared to that before absorbing carbon dioxide, while the viscosity of the upper phase decreased because the carbon dioxide was mainly concentrated in the lower phase. This experimental phenomenon indicates that the upper phase of the absorption system is mainly composed of the organic solvent NPA, while the lower phase is mainly composed of the absorption product and water. A detailed analysis of this aspect was conducted within the reaction mechanisms, thereby substantiating the validity of this result.

The variation of CO_2_ absorption loading with time in absorption systems with different volume ratios of NPA-H_2_O is shown in Figure 3. Within 0–40 min, the absorption load of CO_2_ by the system changes rapidly with increasing time; it can be observed that after 40 min, the system already initiates phase separation, resulting in the formation of a CO_2_-enriched phase with higher viscosity. The absorption capacity of the system for CO_2_ increased gradually over time. In systems with a relatively low water content, phase separation occurs earlier, and the formation of a viscous CO_2_-enriched phase introduces significant mass transfer resistance, thereby impeding the absorption process. For example, a system with an NPA-H_2_O volume ratio of 9:1 reaches absorption saturation at 40 min and exhibits the lowest absorption capacity. Conversely, in systems with a higher proportion of H_2_O, the phase transition process is slower, which is more favorable for CO_2_ absorption. For instance, a system with the NPA-H_2_O volume ratio of 1:1 achieves absorption saturation at 80 min, accompanied by a substantial increase in absorption capacity. At the same time, when the NPA-H_2_O ratio is low, increasing NPA content, the number of active sites available for absorption reactions in the system increases, leading to an increase in absorption loading; however, when the NPA ratio exceeds a certain threshold, excessively high NPA concentrations may induce changes in the system’s physicochemical properties, such as increased viscosity leading to enhanced mass transfer resistance, strengthened hydrogen bonding between NPA molecules reducing their reactivity with the absorbed species, or insufficient water content affecting [TEPAH][Im] and NPA, resulting in the absorption capacity no longer increasing with the ratio and even showing a decreasing trend. Therefore, adjusting the NPA content in the absorbent can effectively enhance its absorption performance. Table 2 summarizes the absorption performance metrics of various ionic liquid-based and phase change absorbents reported in the literature. At absorption temperatures of 303.15 K and 313.15 K, the absorption capacities of other absorbents were all lower than that of [TEPAH][Im]/NPA/H_2_O. The absorption capacities of certain absorbents are even below 1 mol·mol^−1^, and the viscosity of [TEPAH][Im]/NPA/H_2_O remains relatively low. Based on this comparative analysis, the [TEPAH][Im]/NPA/H_2_O absorbent demonstrates superior CO_2_ absorption performance.

#### 2.1.2. The Influence of Temperature on the CO_2_ Absorption Characteristics of [TEPAH][Im]/NPA/H_2_O

In practical applications of absorbents, CO_2_ capture typically occurs subsequent to flue gas desulfurization, with the outlet flue gas temperature fluctuating from 303.15 K to 333.15 K. This temperature interval can substantially influence the phase transition characteristics of the absorbent. This experiment investigated the influence of temperature on CO_2_ capture by [TEPAH][Im]/NPA/H_2_O between 303.15 K and 333.15 K. Figure 4 illustrates the outcomes, while Table 3 presents the physical property metrics of the [TEPAH][Im]/NPA/H_2_O absorbent at saturation under different temperatures. As illustrated in Figure 4, during the start of the reaction, the absorbent maintains a high absorption rate across the different temperature ranges. After 40 min, the absorbents at 323.15 K and 333.15 K reach saturation. After 80 min, the absorbents at 303.15 K and 313.15 K basically reach saturation. As the reaction progresses, the decreasing concentration of [TEPAH][Im]/NPA/H_2_O reduces its reaction with CO_2_, thereby diminishing the absorption efficiency. Upon reaching absorption saturation, the absorption loads of the various absorbents are found to be 1.34, 1.23, 1.11, and 1.05 mol·mol^−1^. When the reaction temperature goes up, the interaction between the absorbent and CO_2_ becomes more intense, causing the absorption rate to speed up. Nevertheless, the reaction of CO_2_ absorption is exothermic, and an increase in temperature is not conducive to the reaction; rising temperatures diminish the solubility of CO_2_ in the solution, giving rise to a consistent reduction in the absorption loading.

### 2.2. Regeneration Performance of [TEPAH][Im]/NPA/H_2_O

#### 2.2.1. Regeneration Efficiency of [TEPAH][Im]/NPA/H_2_O

The regeneration performance of absorbents is a critical criterion for evaluating their effectiveness. According to the proposed reaction mechanism in Section 2.3, the CO_2_ loading absorbed by the phase change absorbent after saturation is mainly concentrated in the lower phase, suggesting that only the lower phase needs regeneration. This study investigates the regeneration efficiency of the ternary absorbent [TEPAH][Im]/NPA/H_2_O under different NPA-H_2_O ratios at 333.15 K and atmospheric pressure, with a desorption time of 60 min. Figure 5 shows that while regeneration at 333.15 K is incomplete with efficiency below 70%, the highest regeneration efficiency occurs at a 1:1 NPA-H_2_O ratio. For a more thorough investigation of the role of desorption temperature plays in the regeneration performance of the [TEPAH][Im]/NPA/H_2_O ternary absorbent, the saturated absorbent was regenerated five times at atmospheric pressures at 333.15 K, 343.15 K, 353.15 K, and 363.15 K. The results can be observed in Figure 6. At a desorption time of 60 min, the regeneration efficiencies at 343.15 K, 353.15 K, and 363.15 K are 78.68%, 84.83%, and 91.08%, respectively, indicating that the regeneration efficiency of [TEPAH][Im]/NPA/H_2_O increases with rising desorption temperature. The underlying reason for this trend is that higher temperatures enhance molecular movement within the solution, destabilizing the products in the saturated solution and facilitating the release of CO_2_ as it breaks bonds and escapes. Taking energy consumption during regeneration into account, continuous temperature increases substantially elevate heating energy consumption, while enhancements in regeneration efficiency may progressively diminish as temperatures rise. At this stage, the cost-effectiveness of the “energy consumption–efficiency” trade-off declines, which in turn reduces its practical applicability. A temperature of 363.15 K is determined to be the most suitable desorption temperature for the [TEPAH][Im]/NPA/H_2_O absorbent. The regeneration performances of various absorbents documented in previous studies are tabulated in Table 4. The regeneration efficiency of other absorbents is all lower than that of [TEPAH][Im]/NPA/H_2_O, and none reach 90%, even at higher temperatures. The findings confirm the outstanding stability of the [TEPAH][Im]/NPA/H_2_O absorbent during regeneration.

#### 2.2.2. Heat Duty for Regeneration of [TEPAH][Im]/NPA/H_2_O

The energy expenditure required for regenerating the saturated absorbent liquid also serves as a vital criterion in determining the applicability of the [TEPAH][Im]/NPA/H_2_O solution as a CO_2_ absorbent [7,36]. The regeneration energy consumption of the absorbent includes the heat of desorption reaction (Q_des_), sensible heat (Q_sen_), and latent heat (Q_latent_). In this study, the vapor–liquid equilibrium (VLE) line for CO_2_ captured using the absorbent was determined within a closed-system reactor. Based on the differences in the gas–liquid equilibrium lines at different temperatures and combined with the Gibbs–Helmholtz equation, the Q_des_ of the solution was further calculated. In order to demonstrate the reliability of the closed reactor system, this study initially measured the VLE line of a 5 M MEA solution at 313.15 K. Subsequently, the acquired results were contrasted with the VLE data of a 5 M MEA solution presented in the existing literature. The VLE data of MEA measured in this paper shows a good overlap with the VLE data reported by Shen et al. [37], which indicates that the closed reactor device built in this paper has good reliability. Subsequently, the VLE data of the [TEPAH][Im]/NPA/H_2_O solution were determined, respectively, at 303.15 K and 313.15 K. The final results are depicted in Figure 7. As the CO_2_ absorption loading of the FILs solution rises, the partial pressure of CO_2_ in the equilibrium gas phase gradually increases. When the temperature inside the closed reactor is elevated from 313.15 K to 323.15 K, the gas-phase partial pressure at equilibrium under the same CO_2_ loading also rises. This behavior can be elucidated by the ideal gas law. Since the volume of the closed reactor stays constant, an increase in temperature leads to a higher partial pressure of the gas phase within the reactor. Based on Equation (4), the average reaction heat per unit mass for CO_2_ is 0.37 GJ·t^−1^CO_2_. The Q_des_ of the MEA solution is 1.77 GJ·t^−1^ CO_2_. Notably, the regeneration heat of this absorbent constitutes merely 21.0% of that of the MEA solution. This is due to the rich-phase product being predominantly [TEPAH]^+^COO^−^ and Im existing in large quantities in the lean solution. Since the rich-phase solid contains almost no [Im]- component during regeneration, the Q_des_ for this absorbing component can be ignored.

In this study, the Q_sen_ of regeneration was estimated based on Formula (18). The Q_sen_ of regeneration of [TEPAH][Im]/NPA/H_2_O is 0.5 GJ·t^−1^CO_2_. The Q_sen_ of this absorbent is only 55.6% of that of the MEA solution; this is because the volume of the regenerant is significantly smaller, and less energy is required in the process of heating the converter to the required temperature. The Q_sen_ data fully reflects the energy-saving benefits of this system. The Q_latent_ of [TEPAH][Im]/NPA/H_2_O solution is 1.33 GJ·t^−1^CO_2_, which is higher than the Q_latent_ value (1.13 GJ·t^−1^CO_2_) of the 30 wt% MEA solution. Lower solute concentration and higher water content in these absorbents explain this phenomenon, as substantial water solvent loss during regeneration elevates Q_latent_.

To summarize, the regeneration energy of the [TEPAH][Im]/NPA/H_2_O absorbent studied in this paper is 2.20 GJ·t^−1^CO_2_. Figure 8 indicates that the ternary phase change absorption system ([TEPAH][Im]/NPA/H_2_O) requires less regeneration energy than the traditional MEA solution. This indicates that the [TEPAH][Im]/NPA/H_2_O phase change absorption system studied in this paper not only has superior absorption capacity and regeneration efficiency but also has a low regeneration energy consumption. Anticipated as an ideal CO_2_ capture absorbent, this material significantly facilitates the advancement of carbon capture technology while demonstrating extensive industrial application potential.

### 2.3. Proposed Mechanism of CO_2_ Capture by [TEPAH][Im]/NPA/H_2_O

#### 2.3.1. Characteristic of CO_2_ Absorption

In an attempt to further examine the reaction process, the absorbent before and after absorption, as well as during, the absorption process was analyzed using carbon-13 nuclear magnetic resonance spectroscopy (^13^C NMR). Figure 9 presents the molecular structures of the reactants that take part in the CO_2_ absorption process of the [TEPAH][Im]/NPA/H_2_O system, as well as the labeling of carbon atoms in the functional groups of the molecular products. In this analysis, carbon atom C1 corresponds to the cation [TEPAH]^+^, while carbon atoms C2, C3, and C3′ represent the anion [Im]^−^. Carbon atom C8 is identified as TEPA-carbamate, C7 corresponds to Im carbamate, and C9 represents the carbon atoms related to HCO_3_^−^/CO_3_^2−^. Figure 10 shows the ^13^C NMR spectra of the solution. In the fresh solution, four signal peaks with chemical shifts of 57.28, 50.74, 47.67, and 39.95 ppm were detected, all of which are assigned to the C1 atom of [TEPAH]^+^. The signal peaks at 135.61 and 121.61 ppm can be ascribed to the carbon atoms in [Im]^−^. Additionally, the peaks at 63.46, 24.51, and 9.44 ppm are assigned to the C4, C5, and C6 carbon atoms in NPA, respectively. After 10 min of absorption, a new cluster of signal peaks appeared in the range of 163–165 ppm, corresponding to the carboxyl carbon of RNCOO^−^. With five amino groups in [TEPAH]^+^, multiple types of carbamates are produced. After 20 min of absorption, a signal peak at 160.1 ppm was identified, corresponding to HCO_3_^−^/CO_3_^2−^. As the absorption reaction proceeded, the intensities of peaks 7 and 8 first increased and then decreased, demonstrating that a hydrolysis reaction of the carbamate occurred in the later period of the rich solution formation. This process leads to the gradual formation of HCO_3_^−^/CO_3_^2−^. Once saturation is achieved, the solution separates into two distinct liquid phases. The upper phase is made up of the poor liquid, and the lower phase is composed of the rich liquid. In the upper solution, strong signal peaks corresponding to NPA and weak signals from [Im]^−^ are observable, but no peaks related to CO_2_ products are detected. As revealed by the ^13^C NMR results, the upper layer of the absorbent consists of a lean-phase solution, primarily composed of NPA. The ^13^C NMR results also indicate that carbamates form in the rich-phase solution. The signal peaks detected at 164.5 and 165.5 ppm correspond to the carboxyl carbons of tertiary and secondary carbamates, while the signal peak at 160.1 ppm corresponds to HCO_3_^−^/CO_3_^2−^. These characterization results reveal that as the reaction progresses, both carbonate and bicarbonate are continuously generated. Due to their high solubility in water and low solubility in NPA, these products concentrate, along with the carbamate, in the lower aqueous phase of the absorbent. In contrast, NPA is primarily concentrated in the upper lean solution, resulting in the observed phase separation. Thus, the majority of the products from CO_2_ absorption accumulate in the rich phase.

#### 2.3.2. Characteristic of CO_2_ Desorption

This experiment further analyzed the distribution of substances during the desorption process using nuclear magnetic resonance (NMR) spectroscopy. The relationship between desorption time and loading capacity was assessed by analyzing solutions at desorption times of 10, 20, 40, and 60 min at 363.15 K. As shown in Figure 11, the signal peak for RNCOO^−^ appears in the range of 164.5–165.5 ppm, with its intensity decreasing as desorption time increases. Similarly, the intensity of the signal peak for HCO_3_^−^/CO_3_^2−^ at 160.3 ppm also gradually diminishes with longer desorption times, and the HCO_3_^−^/CO_3_^2−^ signal peak disappears entirely after a desorption time of 40 min. However, the signal peak for RNCOO^−^ remains detectable even after 60 min of desorption. The NMR spectra indicate that the chemical shifts for the product signal peaks are similar to those observed in both absorption and desorption processes. The inversely correlated peak intensity trends between these two processes directly demonstrate the reversibility of absorption and desorption.

#### 2.3.3. Proposed Mechanism of the CO_2_ Absorption–Desorption Reaction

Using the results presented above, the proposed reaction mechanism for CO_2_ capture by the [TEPAH][Im]/NPA/H_2_O solution has been clarified. As illustrated in Figure 12, the adsorption mechanism of the [TEPAH][Im]/NPA/H_2_O system consists of three core steps that govern its CO_2_ capture behavior: dynamic proton transfer between [TEPAH]^+^ and [Im]^−^ to sustain ionic network stability; nucleophilic attack on CO_2_ by the active amine groups of [TEPAH]^+^, leading to the formation of carbonate intermediates; and temperature-induced phase separation to enrich CO_2_-rich species for efficient regeneration. Each of these interconnected steps inherently imposes specific constraints on the overall adsorption capacity, reflecting a deliberate balance between reactivity and regenerability.

In the initial stage of the reaction, CO_2_ engages with the primary and secondary amines attached to [TEPAH]^+^, as well as with the secondary amines on [Im]^−^. The nitrogen atom of the anion [Im]^−^ reacts in an equimolar ratio with CO_2_ to form imidazole carbamate ([Im]CO_2_^−^), which does not have a deprotonation pathway since the ion lacks a hydrogen atom for deprotonation. Nonetheless, these carbamates are extremely unstable and easily undergo hydrolysis, resulting in the formation of HCO_3_^−^ and CO_3_^2−^, while releasing Im simultaneously. The amines in [TEPAH]^+^ (RNH) can react with CO_2_ to form a zwitterionic intermediate (RNH^+^COO^−^) through a zwitterion mechanism. This intermediate can subsequently react with a proton-accepting base (B) to generate RNCOO^−^. The reaction process can be depicted by means of Equations (1)–(4). Simultaneously, NPA can react with RNCOO^−^ to promote its decomposition in solution. In alkaline conditions, CO_3_^2−^/HCO_3_^−^ reacts with NPA to yield propyl carbonate and H_2_O, with the reverse reaction occurring concurrently, as shown in Equations (5) and (6). As the reaction proceeds, the concentration of [TEPAH][Im] decreases while the CO_2_ loading increases. Eventually, RNCOO^−^ and CO_2_ gradually hydrolyze to produce HCO_3_^−^ and CO_3_^2−^, as represented in Equations (7) and (8).[Im]^−^ + CO_2_ → [Im]CO_2_^−^(1)[Im]CO_2_^−^ + H_2_O → Im+ HCO_3_^−^(2)RNH + CO_2_ → RNH^+^COO^−^(3)RNH^+^COO^−^ + B → BH^+^ + RNCOO^−^(4)RNCOO^−^ + C_3_H_7_OH → RNH + C_3_H_7_OCOO^−^(5)C_3_H_7_OH + HCO_3_^−^ ↔ C_3_H_7_OCOO^−^ + H_2_O(6)RNCOO^−^ + H_2_O → RNH_2_^+^ + CO_3_^2−^(7)CO_2_ + H_2_O → H^+^ + HCO_3_^−^(8)

The desorption reaction fundamentally represents the reverse of the absorption reaction. Based on the ^13^C NMR analysis, the signal peak intensity of RNHCOO^−^ is observed to gradually diminish during the desorption process. This decrease is attributed to the reaction between the protons (H^+^) in the solution and RNHCOO^−^, which leads to the consumption of RNHCOO^−^ and further facilitates the regeneration process. Simultaneously, HCO_3_^−^/CO_3_^2−^ preferentially decomposes under heating since inorganic salt ions decompose more readily than organic carbamate structures under thermal conditions. Additionally, another portion of HCO_3_^−^/CO_3_^2−^ reacts with RNH_2_^+^ to form RNCOO^−^, which subsequently interacts with H^+^ to regenerate RNH and release CO_2_. Consequently, in the desorption ^13^C NMR spectrum, the signal peak of HCO_3_^−^/CO_3_^2−^ weakens preferentially and eventually disappears. The desorption process is represented by Equations (9)–(12).CO_3_^2−^ + 2H^+^ → H_2_O + CO_2_(9)HCO_3_^−^ + H^+^ → H_2_O + CO_2_(10)2HCO_3_^−^ + RNH_2_^+^ → RNCOO^−^ + 2H_2_O + CO_2_(11)RNCOO^−^ + H^+^ → RNH + CO_2_(12)

## 3. Materials and Methods

### 3.1. Materials

Monoethanolamine (MEA, 99%), tetraethylenepentamine (TEPA, 95%), and imidazole (Im, 99%) were all purchased from Shanghai Maclin Biochemical Technology Co., Ltd. (Shanghai, China). N-propanol (NPA, 99.5%) was obtained from Shandong Keyuan Biochemical Co., Ltd. (Laizhou, China). Ethanol (EtOH) was provided by Laiyang City Kant Chemical Co., Ltd. (Laizhou, China). CO_2_ (10.0 v%) and N_2_ (99.9 v%) were produced by Jinan Deyang Special gas Co., Ltd. (Laizhou, China).

### 3.2. Preparation of [TEPAH][Im]/NPA/H_2_O

The synthesis method for the ionic compound involved preparing a 150 mL ethanol–water solution with a volume ratio of 4:1. Equal molar amounts (0.0125 mol) of TEPA and Im were weighed and dissolved in the ethanol–water mixture. The mixture was placed in a single-necked flask and magnetically stirred (DF-101S heat-collecting constant-temperature heating magnetic stirrer, Gongyi Yuhua Instrument Co., Ltd., Gongyi, China) at 25 °C for 24 h to allow sufficient neutralization of acid and base. Subsequently, the mixture was transferred to a round-bottomed flask and completely evaporated using a rotary evaporator (R-1001VNC, Zhengzhou Greatwall Scientific Industrial and Trade Co., Ltd., Zhengzhou, China ) at 70 °C to obtain [TEPAH][Im]. Finally, [TEPAH][Im] was transferred to a volumetric flask, and NPA-H_2_O solutions of various volume fractions were added to a 25 mL volumetric flask for volumetric preparation, resulting in a 0.5 mol·L^−1^ [TEPAH][Im]/NPA/H_2_O solution as the final product.

### 3.3. CO_2_ Absorption–Desorption Experiments

The ionic compound absorbent used in this study had a total volume of 25 mL and a concentration of 0.5 mol·L^−1^. The volume ratios of NPA-H_2_O were varied from 0:1 to 1:0. With the intention of simulating real-world application scenarios, absorption experiments were performed using a 10% CO_2_ concentration at temperatures spanning from 303.15 K to 333.15 K. After the absorption experiments, the volume ratios were analyzed.

The setup illustrated in Figure 13 was used to study the CO_2_ absorption–desorption characteristics and phase change behavior. First, the thermostatic water bath was adjusted to the specified operating temperature. Subsequently, the absorption tube was securely fixed with an iron stand and submerged in the constant-temperature water bath (BHS-2 Digital Thermostat Water Bath, Jiangyin Baoli Scientific Research Instrument Co., Ltd., Wuxi, China). Next, the CO_2_ gas pathway was opened, and its flow rate—regulated by a mass flow controller (MT-52 MFM, HORIBA Precision Instruments (Beijing) Co., Ltd., Beijing, China)—was set to 100 mL·min^−1^. Once the gas flow stabilized, 25 mL of preheated absorbent was poured into the absorption tube to conduct the absorption performance test. A stopwatch and gas analyzer (PTM600-5 Portable Gas Detector, Shenzhen Eranntex Electronics Co., Ltd., Shenzhen, China) were used to record the CO_2_ concentration at the outlet of the absorption tube at various time intervals. After CO_2_ absorption, when two distinct phases formed, they were separated using a separatory funnel for further study. The definition of CO_2_ absorption loading (β, mol·mol^−1^) is as follows:(13)$$\beta =\frac{\left(A_{0}-A\right)\times C\times v}{V_{0}\times n}$$
where A_0_ is the rectangular area (the area of the rectangle with the horizontal coordinate (time) and the vertical coordinate (gas concentration) as the side lengths, respectively, on the coordinate axis); A is the CO_2_ concentration curve integral area; C is the CO_2_ concentration, %; v is the CO_2_ quantity of flow, ml·min^−1^; V_0_ is the standard volume, 22.4 L·mol^−1^; and n is the molar amount of CO_2_, mol.

The desorption process was carried out under normal pressure. A certain amount of CO_2_-saturated solution was added to the bubble absorption tube and maintained at a specified desorption temperature using a thermostatic water bath. Pure N_2_ was introduced into the absorption bottle at a fixed flow rate (100 mL·min^−1^) as a purge gas. A CO_2_ analyzer was used to analyze and record the concentration of CO_2_ in the gas exiting the flask. Desorption was considered complete when the CO_2_ concentration in the gas outlet dropped to 0. The desorbed absorbent was then reused for absorption experiments. After desorption, the absorbent underwent repeated CO_2_ absorption cycles to investigate its regeneration efficiency. The desorption efficiency (η) is calculated as follows:(14)$$\eta =\frac{\;\beta^{\prime }}{{\;\beta }_{0}}\;$$
where β′ is the absorption load of absorbent after regeneration, mol·mol^−1^, and β_0_ is the absorption load of fresh solution, mol·mol^−1^.

### 3.4. Viscosity Analysis

The solution viscosities were determined with a viscometer (capillary inner diameter: 0.5–0.6 mm; viscometer constant: 0.007300 mm^2^/S^2^). Three measurements were performed for each case under normal temperature and pressure. If the readings were reasonably consistent, the average value was taken as the final viscosity.

### 3.5. Nuclear Magnetic Resonance Experiments

^13^C NMR(JEOL Model JNM-ECZL600G, JEOL (BEIJING) CO., LTD., Beijing, China) was utilized to characterize the distribution of substances in the [TEPAH][Im]/NPA/H_2_O solution before and after CO_2_ absorption, as well as during the desorption process(at 10, 20, 40, and 60 min), aiming to elucidate the proposed reaction mechanism of the absorbent for CO_2_ capture. Specifically, 0.5 mL of the absorbent sample and 0.2 mL of D_2_O were carefully injected into a 5 mm diameter NMR tube. Subsequently, ^13^C NMR analysis was performed on the prepared samples.

### 3.6. Heat Duty

To study the thermodynamic characteristics of CO_2_ capture using the [TEPAH][Im]/NPA/H_2_O, this study employed a closed reactor system to determine the vapor–liquid equilibrium data for CO_2_ absorption by this absorbent. The detailed experimental configuration is depicted in Figure 14.

The total energy consumption (Q_regen_) for CO_2_ desorption during the regeneration process of the saturated absorbents consists of three components: desorption reaction heat (Q_des_), sensible heat (Q_sen_), and latent heat (Q_latent_) [39]. The specific calculation formula is as follows:Q_regen_ = Q_des_ + Q_sen_ + Q_latent_(15)

Q_des_ is the heat required to open the CO_2_ absorption products during the regeneration process, which corresponds to the heat emitted during the reaction of CO_2_ with the absorbent and forms CO_2_ absorption bonds during the absorption process [40]. Q_des_ can be expressed as shown in the following equation:Q_des_ = −∆H_abs_(16)

In the formula, under the given load conditions, it can be calculated by the following formula:(17)$$\ln\left(\frac{P_{{\mathrm{CO}}_{2}}^{\mathrm{sat}}\left(T_{1},\alpha \right)}{P_{{\mathrm{CO}}_{2}}^{\mathrm{sat}}\left(T_{2},\alpha \right)}\right)=-\frac{\Delta H_{\mathrm{abs}}}{R}\left(\frac{1}{T_{1}}-\frac{1}{T_{2}}\right)$$
where ΔH_abs_ is the reaction enthalpy, kJ·mol^−1^; Q_des_ is the reaction heat, GJ·t CO_2_^−1^; and $P_{{\mathrm{CO}}_{2}}^{\mathrm{sat}}\left(T_{1},\alpha \right)$ and $P_{{\mathrm{CO}}_{2}}^{\mathrm{sat}}\left(T_{2},\alpha \right)$ refer to the CO_2_ vapor–liquid equilibrium partial pressures (KPa) of the FIL solution at the same CO_2_ absorption load (α, mol·mol^−1^) at different temperatures T_1_ and T_2_ (K).

Q_sen_ refers to the quantity of heat necessary to warm the absorbent’s rich liquid to the temperature demanded for the regeneration process [31]. In this study, the sensible heat of each component in the absorbent can be calculated using the following formula:(18)$$Q_{sen}=\underset{j}{\sum }\frac{C_{p,j}\Delta Tm_{solution,j}}{\Delta m_{CO_{2}}}$$(19)$$C_{p}=\overset{n}{\underset{i=1}{\sum }}\frac{C_{p}^{i}m_{i}}{m_{solution}}$$
where ∆T is the heating temperature difference in the heat exchange tower, which is generally 10 K; ∆$m_{CO_{2}}$ is ${CO}_{2}$ desorption mass at the corresponding time, kg; and $m_{solution}$ is the quality of the absorbent, kg.

Q_latent_ refers to the energy carried away by the gas volatilized by the absorbent during the regeneration process. The specific calculation formula is as follows:(20)$$Q_{latent}=\underset{i}{\sum }\frac{\Delta m_{i}}{\Delta m_{CO_{2}}}H_{i}^{vap}\left(T_{R}\right)\;\;\;\;\;\;$$
where $H_{i}^{vap}(T_{R})$ is the enthalpy of evaporation of component i at the desorption temperature (T_R_), kJ·kg^−1^. The enthalpy of evaporation of water vapor is mainly evaluated by equation:(21)$$\frac{H_{i}^{vap}\left(T_{R}\right)}{H_{i}^{vap}\left(T_{B}\right)}={\left(\frac{1-T_{R}/T_{C}}{1-T_{B}/T_{C}}\right)}^{0.38}$$
where $\;H_{i}^{vap}(T_{R})$ and $H_{i}^{vap}(T_{B})$ are the enthalpy of evaporation of component i at the desorption temperature ($T_{R}$, T_B_), kJ·kg^−1^; T_B_ represents the standard boiling point of the component, K; T_C_ represents the critical boiling point of the component, K; T_R_ represents the set regeneration temperature, K.

## 4. Conclusions

An ionic compound featuring TEPA as the cation and Im as the anion has been successfully synthesized. The [TEPAH][Im]/NPA/H_2_O system is a homogeneous phase prior to CO_2_ absorption but transitions into a liquid–liquid two-phase system afterward. NPA functions as a phase transition accelerator and provides benefits including a fast absorption rate, a small volume of the rich phase, and high regeneration efficiency. The results demonstrate that the new phase change absorbent [TEPAH][Im]/NPA/H_2_O (with a NPA-H_2_O ratio of 1:1) exhibits exceptional absorption performance. At 303.15 K, the absorption loading reaches 1.34 mol·mol^−1^, which is approximately 2.7 times greater than that of 30 wt% MEA. The rich-phase volume accounts for 30%. The viscosity of the absorbent before CO_2_ adsorption is measured at 2.80 mPa·s, while the rich phase viscosity after adsorption increases to 3.58 mPa·s. These values are inferior to those commonly seen in most phase change absorbents, rendering this system favorable for industrial applications. At 363.15 K, the regeneration efficiency is recorded at 91.1%, and this efficiency remains above 70% after five regeneration cycles, indicating a high cyclic absorption capacity and effective regeneration performance. Based on ^13^C NMR analysis, the proposed absorption and desorption mechanism of [TEPAH][Im]/NPA/H_2_O has been elucidated. Both the anions and cations of [TEPAH][Im] can react with CO_2_ to form RNCOO^−^, which is crucial for the absorption performance of the absorbent. The carbamate then reacts with H_2_O and NPA to yield HCO_3_^−^ and C_3_H_7_OCOO^−^. In the NPA-H_2_O mixture, HCO_3_^−^ and C_3_H_7_OCOO^−^ can convert into one another, ultimately achieving equilibrium. Hence, the lower aqueous solution sees the accumulation of high-density CO_2_ products, with H_2_O and NPA predominantly located in the upper lean phase solution. The energy consumption for regeneration of the [TEPAH][Im]/NPA/H_2_O phase change absorbent is only 2.20 GJ·t^−1^CO_2_, which is 41.60% lower than that of the traditional MEA aqueous solution. In conclusion, the [TEPAH][Im]/NPA/H_2_O phase change absorbent has a high absorption capacity and high efficiency. Its regeneration efficiency and low energy consumption make it highly advantageous in industrial promotion.

## Figures and Tables

**Figure 1 molecules-30-03404-f001:**
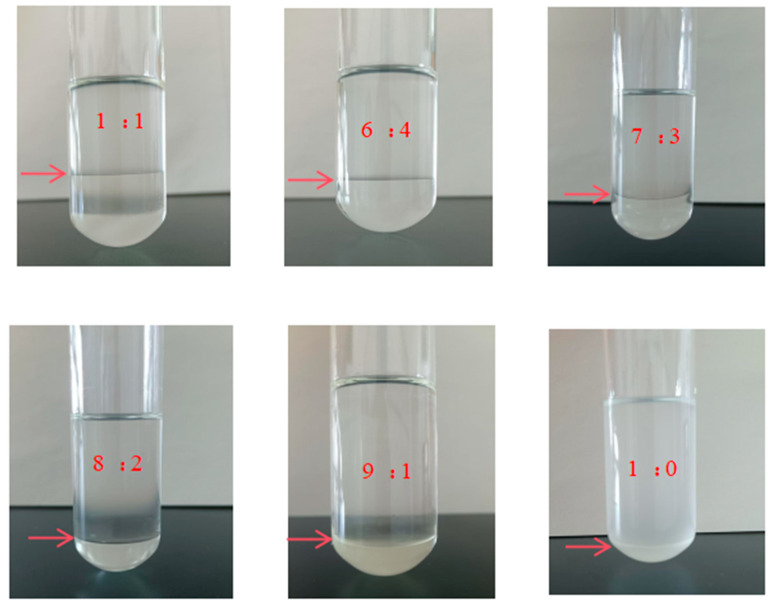
Phase change behavior of [TEPAH][Im]/NPA/H_2_O. (the arrow indicates the boundary between the upper and lower phases formed after phase separation).

**Figure 2 molecules-30-03404-f002:**
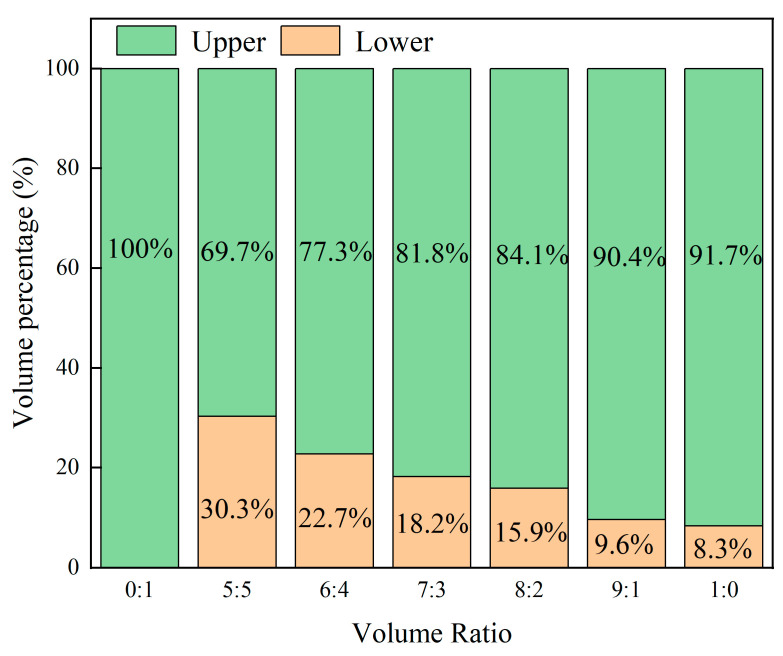
Volume ratio of upper and lower phase.

**Figure 3 molecules-30-03404-f003:**
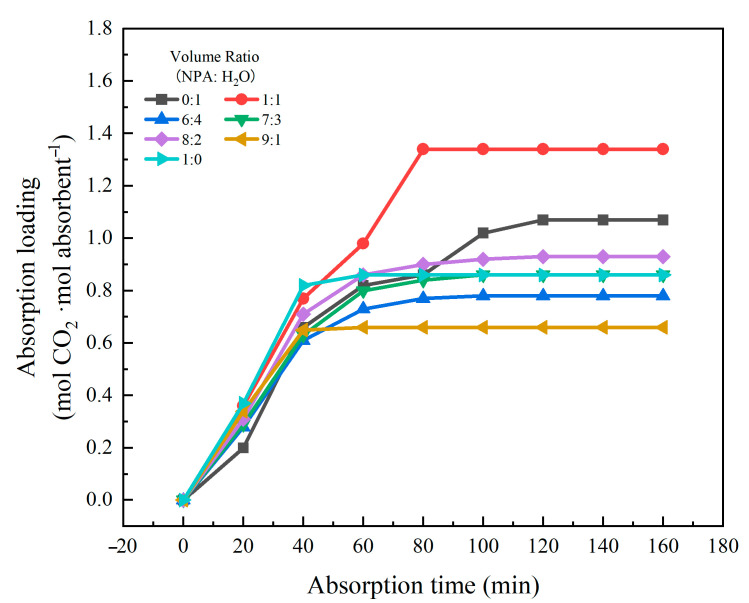
Absorption loading of [TEPAH][Im]/NPA/H_2_O under different solvent volume ratio.

**Figure 4 molecules-30-03404-f004:**
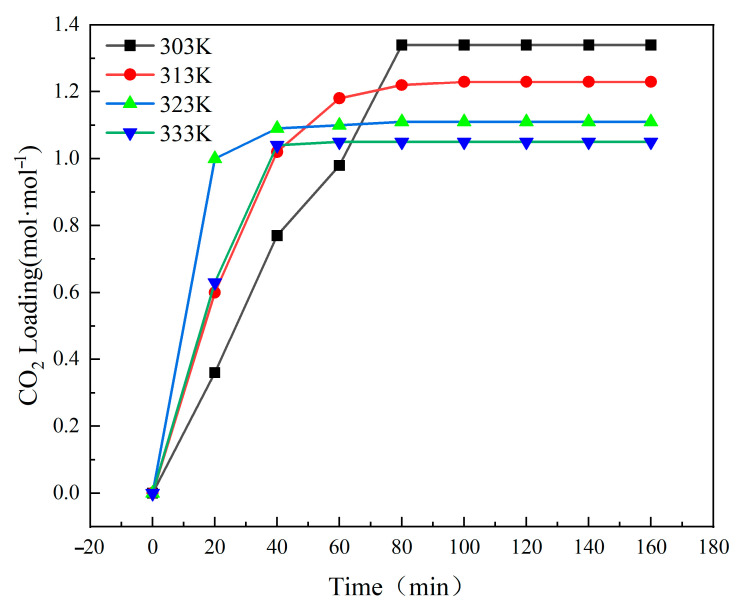
CO_2_ absorption into [TEPAH][Im]/NPA/H_2_O at different temperatures.

**Figure 5 molecules-30-03404-f005:**
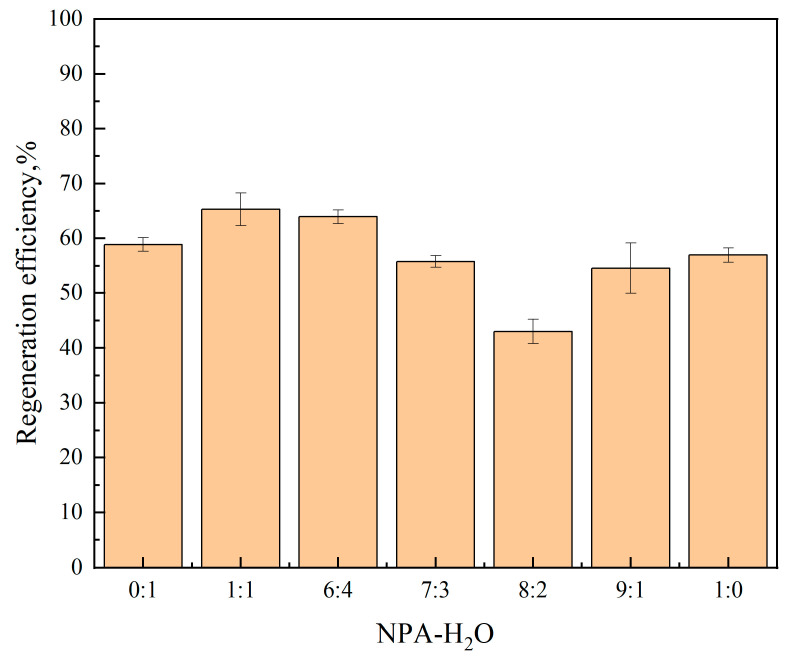
Regeneration efficiency of different NPA-H_2_O ratios.

**Figure 6 molecules-30-03404-f006:**
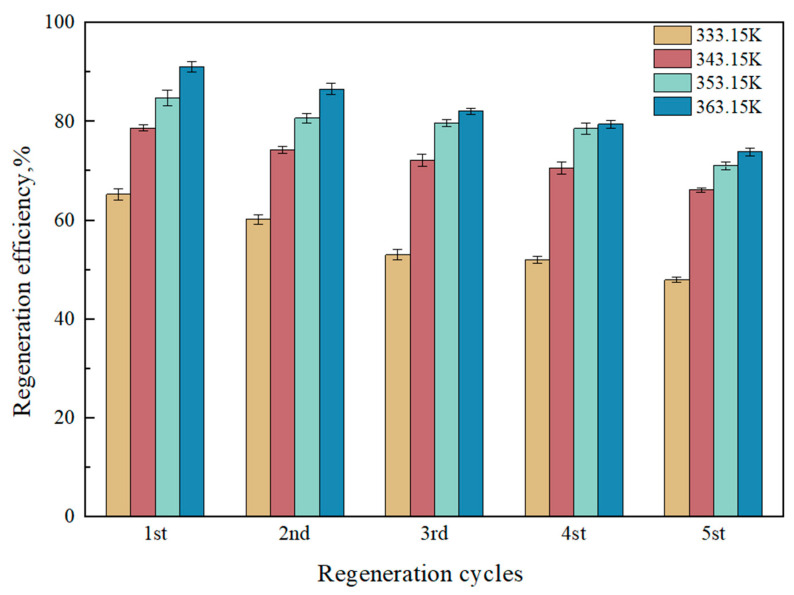
Recycling efficiency at different temperatures.

**Figure 7 molecules-30-03404-f007:**
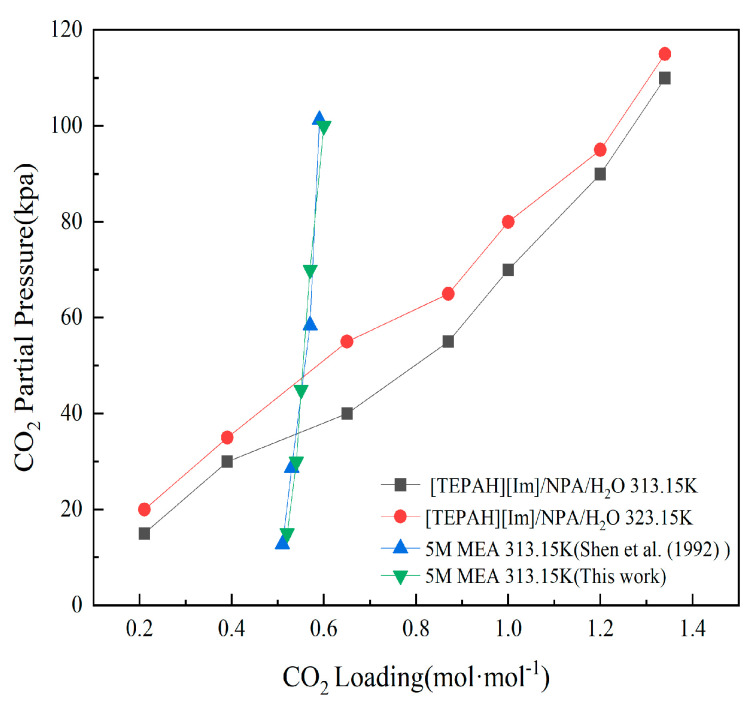
VLE data of [TEPAH][Im]/NPA/H_2_O at 303.15 K and 313.15 K. (the VLE data in the reference is derived from the study by Shen et al. (1992) [37]).

**Figure 8 molecules-30-03404-f008:**
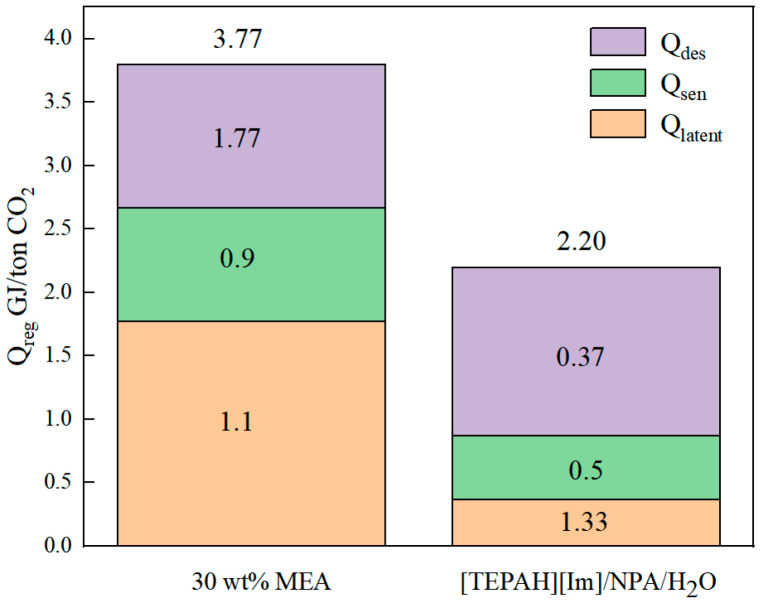
Regeneration energy of the traditional MEA and the [TEPAH][Im]/NPA/H_2_O. The data of MEA were obtained from Wang et al. [38].

**Figure 9 molecules-30-03404-f009:**
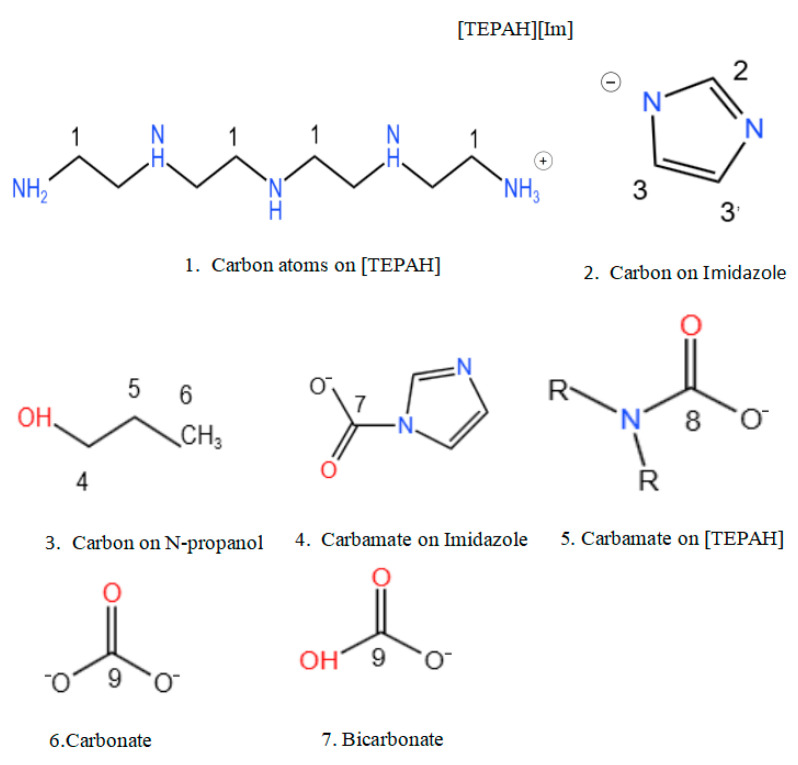
Molecular structure of reactants in [TEPAH][Im]/NPA/H_2_O and molecular type of products in C-atom identity.

**Figure 10 molecules-30-03404-f010:**
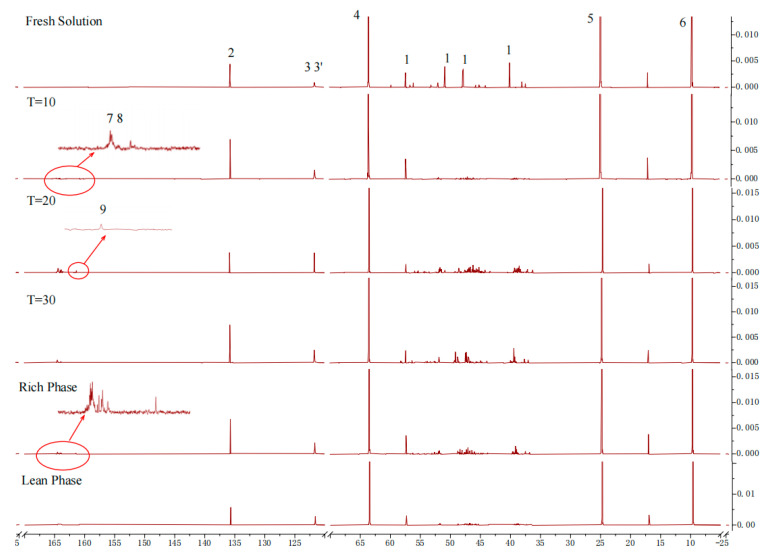
^13^C NMR spectra of the [TEPAH][Im]/NPA/H_2_O.

**Figure 11 molecules-30-03404-f011:**
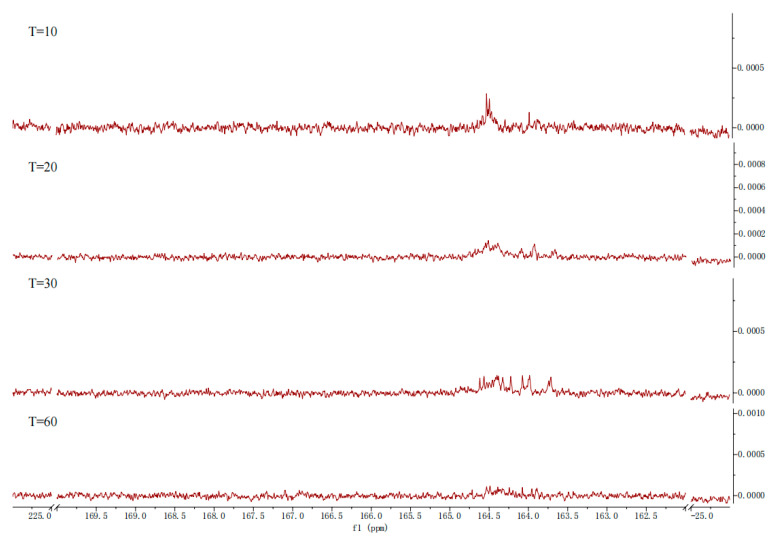
^13^C NMR of [TEPAH][Im]/NPA/H_2_O during the desorption.

**Figure 12 molecules-30-03404-f012:**
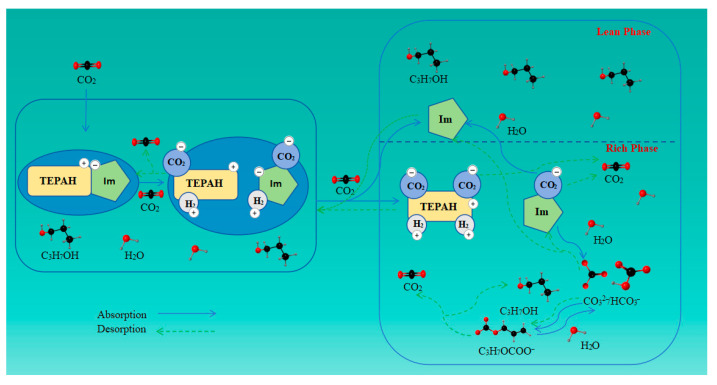
Schematic diagram of the proposed mechanism of CO_2_ capture by [TEPAH][Im]/NPA/H_2_O.

**Figure 13 molecules-30-03404-f013:**
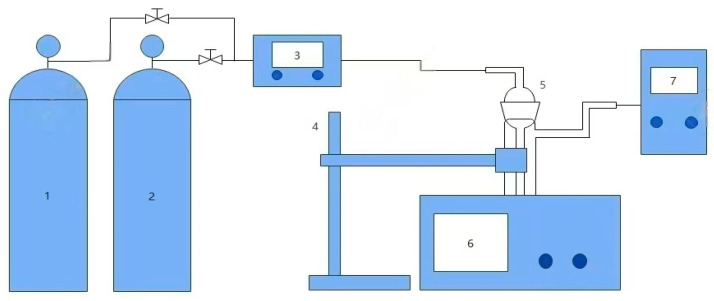
Schematic diagram of the CO_2_ absorption–desorption apparatus. (1. CO_2_ cylinder; 2. N_2_ cylinder; 3. mass flow controller; 4. iron stand; 5. gas absorption bottle; 6. digital thermostatic water bath; 7. gas analyzer).

**Figure 14 molecules-30-03404-f014:**
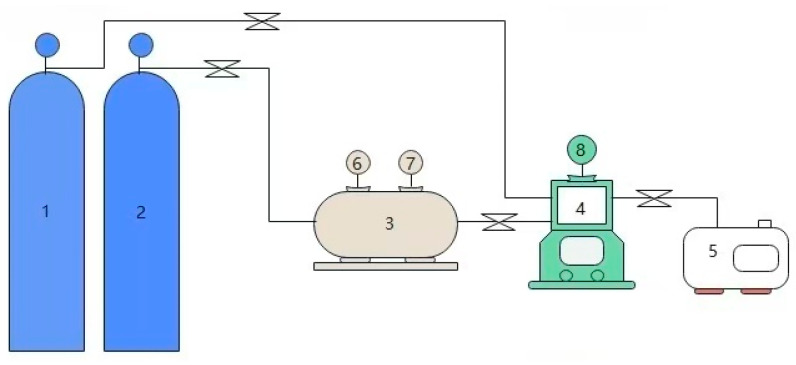
Experimental schematic for setup to determine VLE data (1. N_2_ cylinder; 2. CO_2_ cylinder; 3. gasholder(5L-A); 4. magnetic stirring reactor(GKCF-S); 5. vacuum pump(2XZ-2Rotary disk type); 6, 8. pressure gage; 7. thermometer).

**Table 1 molecules-30-03404-t001:** The properties of [TEPAH][Im]/NPA/H_2_O with different volume ratios of NPA-H_2_O. (C_[TEPAH][Im]/NPA/H_2_O_: 0.5 mol/L; T: 303.15 K; Q_CO_2__: 100 mL/min; V_[TEPAH][Im]/NPA/H_2_O_: 25 mL).

V_NPA_:V_H_2_O_	Phase Separation		Viscosity (mPa·s)		Loading (mol·mol^−1^)
	Before Absorption	After Absorption	Before Absorption	After Absorption	
0:1	N	N	1.23 ± 0.036	1.44 ± 0.029	1.07 ± 0.068
1:9	N	N	1.44 ± 0.056	1.48 ± 0.034	1.03 ± 0.083
2:8	N	N	1.84 ± 0.066	1.88 ± 0.057	1.15 ± 0.069
3:7	N	N	1.91 ± 0.055	1.92 ± 0.044	1.12 ± 0.071
4:6	N	N	2.22 ± 0.039	2.27 ± 0.052	0.95 ± 0.075
1:1	N	Y	2.80 ± 0.048	2.85 ± 0.047 (upper)	1.34 ± 0.094
				3.58 ± 0.044 (lower)	
6:4	N	Y	3.17 ± 0.034	2.81 ± 0.032 (upper)	
				3.70 ± 0.029 (lower)	0.78 ± 0.112
7:3	N	Y	3.28 ± 0.033	2.77 ± 0.027 (upper)	
				3.62 ± 0.031 (lower)	0.86 ± 0.136
8:2	N	Y	3.51 ± 0.042	2.72 ± 0.033 (upper)	
				3.55 ± 0.041(lower)	0.93 ± 0.083
9:1	N	Y	3.63 ± 0.046	2.61 ± 0.044 (upper)	
				3.77 ± 0.046 (lower)	0.66 ± 0.141
1:0	N	Y	4.93 ± 0.035	2.45 ± 0.047 (upper)	
				4.53 ± 0.049 (lower)	0.86 ± 0.106

**Table 2 molecules-30-03404-t002:** Comparison of the absorption performance of different absorbents.

Absorbents	Absorption Loading (mol·mol^−1^)	Saturated Viscosity (mPa·s)	Absorption Temperature (K)	Ref.
EDA/AMP/NMF	1.07	-	303.15	[24]
[IPDAH][Im]/H_2_O	1.26	1.35	313.15	[25]
AMP/PZ/DME	0.87	-	313.15	[26]
AMP/AEEA/NMP	0.74	-	313.15	[27]
MAE/DMSO/PMDETA	0.84	8.87	313.15	[28]
AEP/DEGDME/H_2_O	1.23	6.20	313.15	[29]
[TEPAH][Im]/NPA/H_2_O	1.34	3.58	303.15	This work

(EDA—ethylenediamine; AMP—2-Amino-2-methyl-1-propanol; NMF—methyl formamide; IPDA—isophorondiamine; PZ—piperazine; DME—dimethoxyethane; AEEA—aminoethylethanolamine; MAE—methacrylic acid; DMSO—dimethyl sulfoxide; PMDETA—pentamethyldiethylenetriamine; AEP—N-Aminoethylpiperazine; DEGDME—diethylene glycol dimethyl ether).

**Table 3 molecules-30-03404-t003:** The properties of [TEPAH][Im]/NPA/H_2_O at different temperatures.

Temperature, K	Phase	Viscosity, mPa·s	Loading, mol·mol^−1^
303	upper	2.85 ± 0.055	
	lower	3.42 ± 0.046	1.34 ± 0.094
313	upper	2.66 ± 0.041	
	lower	3.12 ± 0.051	1.23 ± 0.121
323	upper	2.74 ± 0.039	
	lower	3.05 ± 0.044	1.11 ± 0.142
333	upper	2.71 ± 0.041	
	lower	3.42 ± 0.046	1.05 ± 0.088

**Table 4 molecules-30-03404-t004:** Contrasting the regeneration efficiency values of multiple biphasic absorbents.

Phase Change Absorbent	Regeneration Temperature (K)	Regeneration Efficiency	Desorption Time (min)	Ref.
[TEPAH][Im]/NPA/H_2_O	363.15	91.1%	60	This work
MEA/NPA/H_2_O	393.15	67.0%	30	[30]
DEEA/AEEA/H_2_O	373.15	68.4%	120	[31]
TEEA/DEEA/H_2_O	373.15	42.0%	120	[32]
BDA/DEEA/H_2_O	353.15	43.6%	90	[33]
DETA/PMDETA (1:4)/H_2_O	393.15	31.0%	120	[34]
DETA/PMDETA (2:3)/H_2_O	393.15	40.7%	120	[35]

(MEA—monoethanolamine; TEEA—trolamine; DEEA—diethylaminoethanol; BDA—1,4-Diaminobutane; DETA—diethylenetriamine PMDETA—pentamethyldiethylenetriamine).

## Data Availability

The original contributions presented in the study are included in the article; further inquiries can be directed to the corresponding author.

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
