# Peer review of "CO2 Capture Performance and Preliminary Mechanistic Analysis of a Phase Change Absorbent"

_molecules, 2025, doi:10.3390/molecules30163404_

Round 1
Reviewer 1 Report (New Reviewer)
Comments and Suggestions for Authors
Why was a mixture of imidazole and TEPA used? Pure TEPA will have a much larger absorption capacity? Pleas explain.
Why do you believe mixing those gives an ionic liquid?
The 1,34 mol/mol presumption is misleading since you are using a solution of two different absorbents in a mixture of solvents. To which one of the components does the mol ratio refer to?
Fig1 and Fig2 have to be better explained. Is Fig1 before the CO2 absorption or after? What is the composition of the two phases? What is the concentration and ratios of the amines?
How was the absorption performed? Under what conditions. What with? The whole section 2.1.1. has to be rewritten, explaining in detail what and how was done, what is the product, what are the upper and lower phases made of, etc.
How can you regenerate just one of the phases without knowing its composition?
Author Response
Thank you very much for taking the time to review our manuscript entitled "Performance and mechanism of CO2 capture by a novel ionic liquid phase change absorbent" and for providing valuable comments and suggestions. These insights have been instrumental in improving the quality of our work. We have carefully addressed each of your points and revised the manuscript accordingly. Below is a detailed response to your comments, along with explanations of the revisions made. We hope the revised version meets your expectations and kindly request your further review.
The replies are attached.

Reviewer 2 Report (New Reviewer)
Comments and Suggestions for Authors
This manuscript investigates the “Performance and mechanism of CO2 capture by a novel ionic liquid phase change absorbent”. This study aims to develop a promising absorbent for CO2 capture. There are some issues which need to be considered prior to acceptance.
- The authors discussed that CO2 capture, utilization, and storage (CCUS) is regarded as a critical technology for curbing near-term CO2 emissions and has been extensively deployed in industrial settings. I recommend adding recent and most relevant references here [Carbon Capture Science & Technology, 15 (2025) 100405].
- There are various formatting issues. E.g. line # 65 (there should be space between line and reference); line # 68 (there should be no full stop before references).
- Authors present performance of [TEPAH][Im]/NPA/H2O, how about other possibilities. I recommend comparing literature for better understanding.
- Table # 2, the performance is not promising as compared to literature. Could you please elaborate the details?
- There should be one schematic which could describe the overall work
Author Response
Thank you very much for taking the time to review our manuscript entitled "Performance and mechanism of CO2 capture by a novel ionic liquid phase change absorbent" and for providing valuable comments and suggestions. These insights have been instrumental in improving the quality of our work. We have carefully addressed each of your points and revised the manuscript accordingly. Below is a detailed response to your comments, along with explanations of the revisions made. We hope the revised version meets your expectations and kindly request your further review.
The replies are attached.

Reviewer 3 Report (New Reviewer)
Comments and Suggestions for Authors
The authors submitted a manuscript entitled “Performance and mechanism of CO2 capture by a novel ionic liquid phase change absorbent” with the reference molecules-3781725.
The subject of the manuscript is interesting and actual; it is suitable for the magazine and deserve to be analysed. Introduction is a little bit generic and does not state the point with accuracy. The experimental part is generally detailed and well explained. Figures are clear despite in some of them the colours choice not being the best. References seems ok and no sign of over self-citation is detected.
The English of the text is good and understandable, but a revision should be made to correct some gaps of words and typos. (e.g. line 39, “… four main xxxxx comprising capture ….”, where xxxxx is a missing word to make the sentence clear).
However, some aspects must be improved prior to publication:
1 – In Figure 3 the absorption loading values does not change monotonically with the ratio NPA/H2O. This must be explained.
2 – The study seems incomplete without a explanation. They are presented ratios of NPA/H2O of 1:0; 9:1; 8:2; 7:3; 6:4; 1:1 and 0:1 and concluded that the 1:1 is the best one for the desired purpose. And the ratios 4:6; 3:7; 2:8 and 1:9 why are not studied? If the loading is increasing while the NPA/H2O ratio increases until 1:1 how can the authors be sure that it won’t increase even more in the unstudied ratios. The reviewer subjects that the study be completed.
Overall, the manuscript seems a good one and after the flaws been corrected it could be considered for publication.
Comments on the Quality of English Language
Could be improved.
Author Response
Thank you very much for taking the time to review our manuscript entitled "Performance and mechanism of CO2 capture by a novel ionic liquid phase change absorbent" and for providing valuable comments and suggestions. These insights have been instrumental in improving the quality of our work. We have carefully addressed each of your points and revised the manuscript accordingly. Below is a detailed response to your comments, along with explanations of the revisions made. We hope the revised version meets your expectations and kindly request your further review.
The replies are attached.

Reviewer 4 Report (New Reviewer)
Comments and Suggestions for Authors
Overall the work is on point, well simplified and relevant for current needs CO2 capture needs that are relevant to climate change concerns.
The title is misleading. It is overly ambitious to claim novelty in what already exists or a mechanism that is not fully studies and confirmed through experimental evidence. Title must be revised accordingly or additional data needs to be provided to confirm claims.
Table 10: The C13 spectrum for the lean phase is not scaled well relative to the others yet there is an impression that some peaks are missing from the spectrum. Scale to relative size. Additionally, full spectra must be provided, inserts with zoomed spectra can then be used to highlight certain peaks of interest. Zoomed spectra can also be given separately and then full spectra provided in supplementary. The way the x axis appears and the way the spectra are presented makes it look like there was a lot of tampering with the spectra. It might be important to supplement the mechanistic claims with additional data from instruments such as MS.
Thermodynamics and kinetics calculations that accompany the data can be provided to provide more insight into the absorption mechanism and predicting optimal absorption/desorption conditions.
There are no reasons why absorbent can not be fully regenerated to 100% and why efficiency decreases over cycles.
The CO2 absorption/desorption mechanism must be presented as a 'proposed mechanism' otherwise supporting experimental data will be required.
A few minor errors in writing were highlighted.

Author Response
Thank you very much for taking the time to review our manuscript entitled "Performance and mechanism of CO2 capture by a novel ionic liquid phase change absorbent" and for providing valuable comments and suggestions. These insights have been instrumental in improving the quality of our work. We have carefully addressed each of your points and revised the manuscript accordingly. Below is a detailed response to your comments, along with explanations of the revisions made. We hope the revised version meets your expectations and kindly request your further review.
The replies are attached.

Round 2
Reviewer 1 Report (New Reviewer)
Comments and Suggestions for Authors
Thank you for all the hard work on the manuscript.
Due to the low pKa difference between imidazole and TEPA this is not an ionic liquid, just a mixture. I have been working with ILs for more than 20 years, I am perfectly aware what an IL is.
Your explaination do not rationalize the use of those components. Why not take Methylimidazole, it is even more liquid than imidazole?
1,34mol/mol is a very low value, since the theoretical capacity of pure TEPA is above 4 mol/mol so it can not possible this to be for the molar concentration of the TEPA. I am not sure you understood my question.
The absoprion mechanisim is plausible, but does not explain the low sorption capacity
The tables are much better readable, thank you.
Author Response
Please find attached:

Reviewer 2 Report (New Reviewer)
Comments and Suggestions for Authors
The revision is acceptable
Author Response
Please find attached:

Reviewer 3 Report (New Reviewer)
Comments and Suggestions for Authors
Ther authors improve a lot the manuscript. The only thing it should be already improved is to introduce in Figure 1 or 2 the values for the experimental data for NPA/Hâ‚‚O ratios of 4:6,
3:7, 2:8, and 1:9. In their letter the authors state they have measured the values for these gradients, so why not show them?
Author Response
Please find attached:

Reviewer 4 Report (New Reviewer)
Comments and Suggestions for Authors
Thank you for addressing all the issues of concern. In the future you might need to use reference standards if peak changes require tight monitoring.
Author Response
Please find attached:

This manuscript is a resubmission of an earlier submission. The following is a list of the peer review reports and author responses from that submission.
Round 1
Reviewer 1 Report
Comments and Suggestions for Authors
The submitted manuscript report on the performance of IL phase change absorbents for CO2 capture. The manuscript has serious flaws and cannot be accepted for publishing in Molecules in the current form. Here is a list of some comments:
RESULTS AND DISCUSSION:
- adsorption is not the same as absorption; the two terms are used interchangeably
- In both Figure 1 and 2 it needs to be explained the volume ratio of what is presented? Are the values given in both Figure 1 and 2 for different scenarios – one, for the ratio of liquid phases (IL:alcohol/water) and another, for different ratio of alcohol and water (alcohol:water)? Or is it all the same: always ratio of alcohol to water (alcohol:water)? If it is always alcohol:water ratio, how come the phase split is not visible in the photos? Maybe some arrow to indicate the phases’ border? The information given is not explicit and it makes the analysis difficult.
- 5:5 is simply 1:1, right?
- It is necessary to depict (in figure 1 or in some schematic drawing) which phase is rich in what? Kind of a graphic representation of what is further described in part 2.3.2. to facilitate the understanding of the content of the manuscript. Where tend to remain/go the components of the system after absorption experiment? What is the composition of each phase at given moment?
- Suggestion is to change the legend in Figure 2 – it is more intuitive and easier to read if the lower phase (here green) will be on the bottom of the bar, and upper phase (here orange) will be on the top, representing real situation in the flask;
- All data are missing uncertainties (Tables)
- Line 107: However, in the range of 5:5 to 8:2, the absorption load first rises and then falls. – why? What could be the reason for that? Please elaborate
- Lines 110/113: meaningless explanation
- Lines116-117: The viscosity of the rich phase shows no significant difference between its state before absorption and after achieving absorption saturation. – Rich in what? Wasn’t the lower phase absorbing the CO2? The upper phase is with an unchanged viscosity…
- Lines 133-134: Figure 4, not 3
- Line 138: After 30 minutes, the absorption rates at 323.15 K and 333.15 K began to decrease significantly, while at 303.15 K and 313.15 K, a noticeable decline in absorption rates was also observed. – not true, it simply reached plateau after 40 minutes and approximately 80 minutes, respectively
- . the absorption rate of the absorbent decreased markedly – this is an incorrect conclusion! It simply reached saturation
- The regeneration performances of various absorbents documented in previous studies are tabulated in Table 4. – again, for how many cycles, at which conditions?
- For practical reasons (industrial applications), it is always advisable to represent the solubility of CO2 also in molality (moles of CO2 per amount/mass of solvent), particularly for “big” ionic liquids
- Why not attempt regeneration under vacuum?
- Figure 7 should include the reference for the literature data (which source it was taken from?)
- At which conditions were experiments performed for the MEA solution (Figure 8)? This article (Wang et al.) itself contains more systems calling for comparison
- Overall, the results are poorly discussed (limited to plain description of what can be seen directly from a given figure, table) and not properly compared with the existing (plentiful) literature; Simple statements as: Based on this data, the [TEPAH][Im]/NPA/H2O absorbent demonstrates excellent absorption performance. OR The regeneration performances of various absorbents …are tabulated in Table 4. The findings confirm the outstanding stability of the [TEPAH][Im]/NPA/Hâ‚‚O absorbent during regeneration are NOT enough! More information should be extracted, i.e., at which operational conditions, what are the differences and similarities, at which parameters (if at all) exists the phase split, which system is more environmentally friendly (green principles, cost, availability/synthesis method), what about regeneration characteristics/conditions? Such a comparison is necessary to present the impact and give perspective.
- all figures and tables, together with a corresponding caption, should stand on their own and provide all the necessary information without the need of consulting the main body of the text – that is not the case in this manuscript
MATERIALS AND METHODS
- Line 363, page 17: Once the absorption experiments were completed, the volume ratios were analyzed. – How? What is the “gas analyzer”? If it was done by gas chromatography, please provide full description of the equipment and conditions
- Line 364 (page 18): the setup is illustrated in Figure 13, not 12; also, which equipment exactly was used (brand, model, precision, etc. of thermostatic water bath, mass flow controller, equipment to preheat absorbent, stopwatch, analyser…)?
- - Line 376: which rectangular area, which concentration curve? Please elaborate on the method of analysis for better understanding!
- the sentence in line 379 (page 18): The schematic illustration of the desorption device is depicted in Figure 13. is unnecessary (repetition of information given in line 364)
- How was the desorption performed? If simply flushing with N2, please specify at which conditions?
- Line 390, page 19: which type of viscometer? Brand, model, precision? Which conditions?
- Line 396: how the NMR studies were performed DURING the desorption process? Was it also done DURING the absorption process, or only BEFORE and AFTER? Not clear, also, again, which equipment and conditions were used?
- Heat duty studies: again, equipment (brand, model, precision) and conditions? Line 440 is missing Hivap symbol, and all the T symbols (TB, TC, TR) are missing unit
- In general, lack of information on the types and/or accuracies of the measuring devices, plus the uncertainty and error analysis is not reported
- The manuscript must be checked for numerous typographical and grammatical errors; also, some information is repeated in many places; please, be more concise
- References incorrectly formatted
The manuscript must be checked for numerous typographical and grammatical errors; also, some information is repeated in many places; please, be more concise.
Reviewer 2 Report
Comments and Suggestions for Authors
Comments from Reviewer
Performance and mechanism of CO2 capture by a novel ionic liquid phase change adsorbent
The current form's presentation of methods and scientific results is satisfactory for publication in the Molecules journal. Some comments apply to the entire article. Please take this into account when making corrections. The minor and significant drawbacks to be addressed can be specified as follows:
Minor comments:
- Please do not use abbreviations in the abstract if not necessary. For example, ILPS and AAIL.
- Literature should also be standardized: the size of letters in the titles of journals, the initials of names, and the size of letters in the titles of articles. References require further formatting, modification, and refinement. Check the format of references. The journal name needs to be abbreviated. Some references have incomplete information. See, for example, ref. 31.
Major comments:
- I am not entirely convinced that the authors use the terms adsorption and absorption with understanding. An example is the text found in lines 85 and 86. A good solution is to introduce the concept of sorption.
- Fig. 3. How many times was the measurement repeated? What is the measurement error?
- Fig. 3. Is there a correlation between a and b for absorption times higher than 80 min? Does the data collected in this table refer to the same temperature?
- Tab. 2. Does the data collected in this table refer to the same temperature?
- I'm not entirely convinced that chapter 3.5. Heat duty needs to be so extensive.
Sincerely,
The reviewer.
Reviewer 3 Report
Comments and Suggestions for Authors
After carefully reviewing the manuscript " Performance and mechanism of CO2 capture by a novel ionic liquid phase change adsorbent", I recommend major revisions and/or explanations from the authors to ensure clarity and coherence.
I have provided detailed feedback and suggestions for improvement in the review below.
- There are multiple grammatical issues and awkward phrasing. For example, in the Introduction section the word “regarged” is used. Also, the missing articles, incorrect prepositions should be corrected.
- Figures are blurry, low-resolution, and poorly formatted.
- The manuscript's frequent and repetitive shifts between figures, tables, and disconnected text—particularly in sections 2.1 and 2.2—result in a fragmented and disjointed presentation lacking smooth transitions or cohesive discussion.
- I would suggest placing Figure 9 (chemical structures) at the beginning of the results and discussion section. In current form the reader must go down below to see them. Also, the full name of [TEPAH][Im] is missing. It should be place in the first place this abbreviation is mentioned.
- In Table 2 and 4, there are multiple abbreviations, such as EDA, NMF. AMP etc. They should be explained below the table, because not everyone who will read this will be familiar with them. The Authors need to review this entire manuscript and explain all the abbreviations that occur.
- In the Materials and Methods section, there is no section about the synthetic procedure for the synthesis of ionic liquid. The Authors report only preparation of a dilution that was later used for CO2 Have the Authors obtained pure IL and characterize it using 1H NMR? Was the purity assessed, or just equimolar amounts of starting materials were mixed?
- The Authors should explicitly define units for each variable in equations, and convert all masses consistently (e.g., all to kg or g).
- The desorption time intervals (10, 20, 40, 60 min) are noted, but no temperatures are listed in this section. Desorption behavior depends heavily on temperature, so this should be noted in section 2.3.2.
- All reported values (especially in Figure 8) are single-point estimates with no uncertainty, error bars, or sensitivity analysis. For thermodynamic properties derived from experimental VLE, this is problematic.
- In Table 4 the values are drawn from multiple literature sources, but experimental conditions may differ significantly, this information should be added.
- Revise sentences throughout for grammar and clarity.
The English language throughout the manuscript requires significant improvement for clarity and readability. Grammatical errors, awkward phrasing, and inconsistent terminology hinder comprehension and must be addressed before the manuscript can be considered for publication.